



# Integration of the Vegetation Phenology Module Improves Ecohydrological Simulation by the SWAT-Carbon Model

Mingwei Li[1], Shouzhi Chen[1], Fanghua Hao[1], Nan Wang[1], Zhaofei Wu[1], Yue Xu[2], Jing Zhang[1], Yongqiang Zhang[3], Yongshuo H. Fu[1]

[1]College of Water Sciences, Beijing Normal University, Beijing 100875, China
[2]College of Urban and Environmental Sciences, Central China Normal University, Wuhan 430079, China
[3]Key Laboratory of Water Cycle and Related Land Surface Processes, Institute of Geographic Sciences and Natural Resources Research, Chinese Academy of Sciences, Beijing 100101, China

*Correspondence to*: Yongshuo H. Fu (yfu@bnu.edu.cn)

**Abstract.** Vegetation phenology and hydrological cycles are closely interacted from leaf and species levels to watershed and global scales. As one of the most sensitive biological indicators of climate change, plant phenology is essential to be simulated accurately in hydrological models. Despite the Soil and Water Assessment Tool (SWAT) has been widely used for estimating hydrological cycles, its lack of integration with the phenology module has led to substantial uncertainties. In this study, we developed a process-based vegetation phenology module and coupled it with the SWAT-Carbon model to
investigate the effects of vegetation dynamics on runoff in the upper reaches of Jinsha River watershed in China. The modified SWAT-Carbon model showed reasonable performance in phenology simulation, with root mean square error (RMSE) of 9.89 days for the start-of-season (SOS) and 7.51 days for the end-of-season (EOS). Simulations of both vegetation dynamics and runoff were also substantially improved compared to the original model. Specifically, the simulation of leaf area index significantly improved with the coefficient of determination ($R^2$) increased by 0.62, the Nash–
Sutcliffe efficiency (NSE) increased by 2.45, and the absolute percent bias (PBIAS) decreased by 69.0 % on average. Additionally, daily runoff simulation also showed notably improvement, particularly noticeable in June and October, with $R^2$ rising by 0.22 and NSE rising by 0.43 on average. Our findings highlight the importance of integrating vegetation phenology into hydrological models to enhance modeling performance.

## 1 Introduction

Vegetation plays a crucial role as a link between the land and the atmosphere through its influence on the processes of the carbon, water, and energy cycles (Bonan, 2008; Zhou et al., 2023). Plant phenology is one of the most sensitive bioindicators of climate change (IPCC, 2021; Fu et al., 2015; Vitasse et al., 2022), closely interacting with the hydrological cycles (Chen et al., 2022a; Hwang et al., 2014; Kim et al., 2018). Therefore, vegetation phenological processes must be accurately simulated accurately in hydrological models. The Soil and Water Assessment Tool (SWAT) has been widely used to model
watershed hydrological characteristics (Bhatta et al., 2019; Tian et al., 2020). However, the performance of its vegetation





module and its simulation of vegetation phenology remains poor, resulting in large uncertainties in the simulation of hydrological processes (Chen et al., 2023). To improve predictions of ecohydrological processes under future climate change conditions, it is essential to modify the vegetation module by considering vegetation phenology.

In recent decades, climate change has substantially shifted vegetation phenology, with an advance in the start-of-season
(SOS) (Fu et al., 2014) and a delay in the end-of-season (EOS) (Piao et al., 2019). This has contributed to the extension of the growing season by 2–10 days per decade (Zhao et al., 2015; Garonna et al., 2016; Shen et al., 2022), closely interacting with hydrological processes from leaf, species levels to watershed and global scales (Chen et al., 2022b). At the leaf scale, earlier spring leaf-out would enhance the transpiration and increase the water requirement of plants, resulting in increased water flux from the soil to the atmosphere. At the watershed scale, the extension of the growing season would reduce runoff
through increased evapotranspiration (ET) (Gaertner et al., 2019; Geng et al., 2020; Kim et al., 2018). Shifting in vegetation phenology affects both the magnitude and the timing of hydrological processes (Creed et al., 2015; Hwang et al., 2023). Therefore, developing a hydrological model capable of characterizing vegetation phenology is essential for comprehensive understanding hydrological processes.

The SWAT has been extensively used to assess the impact of climate change on water processes across a variety of
watershed scales. The updated SWAT-Carbon model was developed to improve performance in terms of terrestrial carbon and water cycles (Mukundan et al., 2023; Zhang et al., 2013). As in the original version, SWAT-Carbon employs a simplified version of the Environmental Policy Impact Climate (EPIC) vegetation growth module (Arnold et al., 1998), which determines SOS and EOS based on a daylength threshold calculated as a function of latitude. However, phenological processes are complex, determined not only by daylength, but also by temperature and precipitation (Piao et al., 2015; Wu et
al., 2023; Peñuelas et al., 2004). For example, temperature is regarded as the dominant driver of spring phenology, with spring warming largely advancing leaf-out dates by meeting the forcing requirement earlier (Ge et al., 2015; Fu et al., 2018). However, winter warming-induced reductions in chilling accumulation would increase the forcing requirement and delay the leaf-out date (Fu et al., 2015; Wu et al., 2023). Therefore, the EPIC module cannot capture the real phenological processes (Ma et al., 2019), and coupling specific phenology module with the SWAT-Carbon model is urgently needed to accurately
simulate hydrological processes.

The Tibetan Plateau has experienced seriously climate change, greatly influencing vegetation dynamics and regional water cycles (Shen et al., 2022; Li et al., 2022). In this study, utilizing a typical watershed in the Tibetan Plateau, upper reaches of the Jinsha River watershed, we first developed a process-based vegetation phenology module consisted of spring and autumn phenology models that considers both temperature and photoperiod. Subsequently, we coupled this module into the SWAT-
Carbon model. The objectives of our study are as follows: 1) to develop a process-based vegetation phenology module and integrate it into the SWAT-Carbon model; 2) to elucidate the effects of vegetation phenology and climate change on





watershed ecohydrological processes; and 3) to forecast future shifts in runoff under different emission scenarios using the modified SWAT-Carbon model.

## 2 Materials and Methods

### 2.1 Study Area

Our study focused on the upper reaches of the Jinsha River watershed in China, and the Jinsha River is located in the upper reach of the Yangtze River (Figure 1a). The elevation difference of the upper Jinsha River watershed is 4,773 m (Figure 1b), and the drainage area is 216,440 km$^2$. The multiyear mean temperature is approximately -2.56 ± 4.11 °C and the multiyear mean annual precipitation is approximately 467 ± 135 mm from 1982 to 2018. Influenced by monsoons, precipitation is

mainly observed during June–September, accounting for over 75 % of the annual total, with peak precipitation typically occurring in July. The land use types in the study area showed minimal change during 2007–2014, with grassland (61.4 %) and forest (14.6 %) representing the dominant vegetation types (Figure 1c). Grassland areas are primarily distributed in the upstream region of the basin, whereas forests are prevalent in the downstream area.

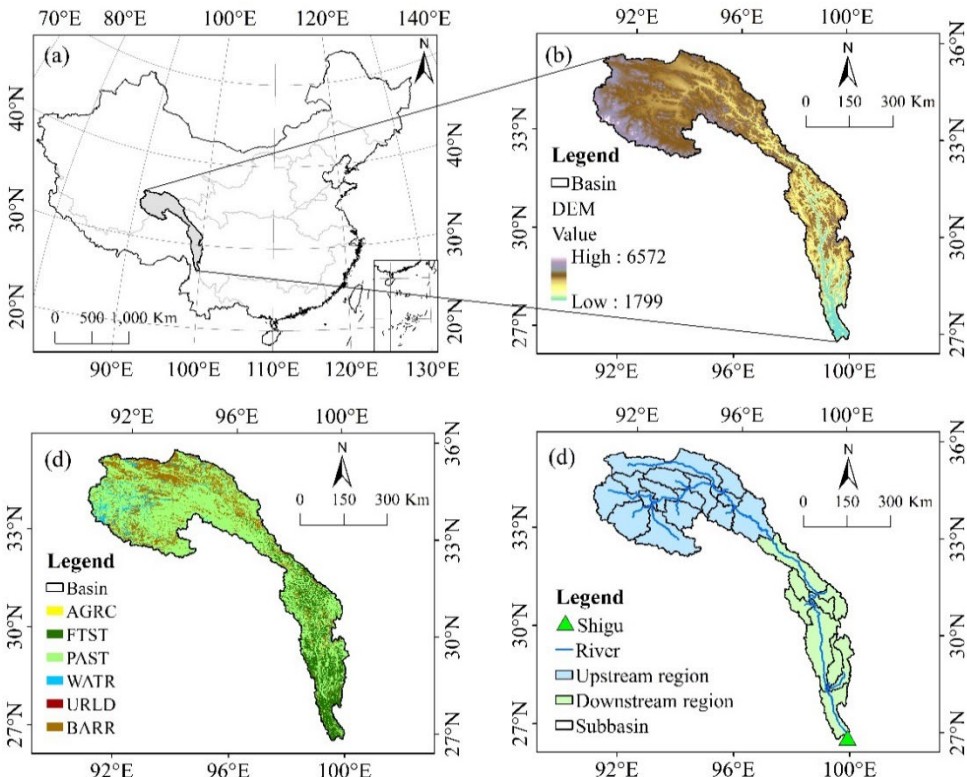

**Figure 1: Study area.** (a) Location of the upper Jinsha River watershed, (b) digital elevation model (DEM), (c) distribution of land use types (AGRC-farmland, FRST-forest, PAST-grassland, WATR-water, URLD- residential area, BARR-unused land), and (d) division of subbasins and the location of the Shigu hydrological gauging station.





## 2.2 Dataset

The sources of the datasets used for phenology models and the SWAT-Carbon model are listed in Table 1.

### 2.2.1 Digital Elevation Model (DEM), Land Use, and Soil Data

The watershed and river network of study area were delineated using a digital elevation model (DEM) with 90-m resolution (Figure 1b) in ArcSWAT2012. The watershed was first subdivided into 28 subbasins (Figure 1d) using topographic characteristics, and then discretized into 1922 hydrologic response units (HRUs) based on land use, soil data, and slope characteristics.

We extracted land use data with 1-km spatial resolution from the Resources and Environmental Sciences and Data Center, Institute of Geographic Sciences and Natural Resources Research, Chinese Academy of Sciences (https://www.resdc.cn/). This extracted data were reclassified to match the land use classes for HRU delineation in the SWAT model (Figure 1c). There were six classes of land use identified in the watershed: farmland (ARGC), forest (FRST), grassland (PAST), water (WATR), residential area (URLD), and unused land (BARR).

The soil map of the upper reaches of the Jinsha River watershed was obtained from the Food and Agriculture Organization of the United Nations (http://www.fao.org). We used the Soil Water Atmosphere Plant model (SWAP: http://hydrolab.arsusda.gov/soilwater) to develop the database of soil properties for the studied watershed.

### 2.2.2 Climate Data and Runoff

Because of the limited distribution of meteorological stations in the study area, gridded climate data were used as observed values to force both the original and the modified SWAT-Carbon models. We used daily meteorological data from 1982 to 2018 with 0.1° spatial resolution extracted from the China Meteorological Forcing Dataset (CMFD), which included precipitation, temperature, relative humidity, wind speed, and solar radiation (He et al., 2020). This dataset was developed by the Institute of Tibetan Plateau Research under the Chinese Academy of Sciences, and it represents the optimal combination of in situ observations, satellite retrials, and meteorological reanalysis.

We selected four CMIP6 models, i.e., CanESM5, FGOALS-g3, MPI-ESM1-2-HR, and MRI-ESM2-0 under three emission scenarios (SSP1-2.6, SSP2-4.5 and SSP5-8.5), to predict vegetation phenology and future runoff. Daily time series of mean temperature, maximum temperature, minimum temperature, and precipitation were acquired from the CMIP6 website (https://esgf-node.llnl.gov/search/cmip6/). CMIP6 data are biased in local areas (Stevens and Bony, 2013) and therefore it is necessary to correct those errors. First, we interpolated the CMIP6 data to 0.1° spatial resolution using linear interpolation. Then, the CMFD was employed as the reference to correct the bias in CMIP6 models using the empirical mapping technique



(Gudmundsson et al., 2012; Wilcke et al., 2013). Daily empirical cumulative distribution functions for the calibration period were produced using a moving window of ±15 days (Thrasher et al., 2012).

Daily runoff data (2007–2014) of the Shigu hydrological gauging station, located at the upper reaches of Jinsha River watershed outlet, were obtained from the Chinese Hydrological Data Yearbook.

**2.2.3 Vegetation Data**

We obtained the Global Inventory Modeling and Mapping Studies third generation normalized difference vegetation index (GIMMS3g NDVI) dataset from the Ecological Forecasting Lab (https://data.tpdc.ac.cn/en/data/9775f2b4-7370-4e5e-a537-3482c9a83d88/) for 1982–2020 (spatial resolution: 1/12°, temporal resolution: every 15 d).

Leaf area index (LAI) data from 1982 to 2018 was acquired from the Global Land Surface Satellite (GLASS) product with 0.05° spatial resolution and an eight-day temporal interval (http://glass-product.bnu.edu.cn). LAI time series for forest and grassland areas were obtained through area-weighted averaging of FRST and PAST grid values within the studied watershed (Strauch and Volk, 2013; Zhang et al., 2020).

**Table 1: Details of the data used in this study.**

| Data | Source | Characteristics |
|---|---|---|
| Digital elevation model (DEM) | Geospatial Data Cloud (https://www.gscloud.cn/) | SRTMDEM 90M |
| Land use types | Center for Resources and Environmental Sciences and Data (https://www.resdc.cn/) | 1 km spatial resolution |
| Soil map | Food and Agriculture Organization of the United Nations (http://www.fao.org) | Harmonized World Soil Database v 1.2 |
| Observed meteorological data | China Meteorological Forcing Dataset (https://data.tpdc.ac.cn/home) | daily, 0.1° spatial resolution |
| Observed runoff | Chinese Hydrological Data Yearbook | Shigu hydrological gauging station, daily |
| Normalized difference vegetation index (NDVI) | Global Inventory Modeling and Mapping Studies third generation (GIMMS$_{3g}$) (https://data.tpdc.ac.cn/en/data/9775f2b4-7370-4e5e-a537-3482c9a83d88/) | 15 d, 1/12° spatial resolution |
| Leaf area index data (LAI) | Global Land Surface Satellite (GLASS) product (http://glass-product.bnu.edu.cn). | 8 d, 0.05° spatial resolution |


**2.3 SWAT-Carbon Model Modification**

The SWAT is a comprehensive, process-oriented, and physically based/process-based semi-distributed hydrological model (Neitsch et al., 2011). The SWAT-Carbon model (https://sites.google.com/view/swat-carbon/home), as an updated vision, is developed to enhance the simulation of terrestrial carbon cycles. However, the SWAT-Carbon model still performs poorly in estimating vegetation dynamics (e.g., phenology and LAI). More realistic phenology data could enhance the performance of LAI simulation, which would further improve the accuracy of runoff simulated by the hydrological model. Therefore, we

developed a novel method to enhance the SWAT-Carbon model by integrating a process-based vegetation phenology module into it. This phenology module consists of two phenology models: the UniChill model for simulating SOS and the DM model for simulating EOS. These phenology models provide dynamic simulation of plant phenology to replace the exact dates of the SOS and EOS in SWAT-Carbon model without the need for any management settings in the operation schedule.

The UniChill model, proposed by Chuine (2000), hypothesizes that spring phenology is regulated both by chilling

temperatures and by forcing temperatures during the dormancy period. Because of the certain range of chilling and forcing temperatures, we added temperature conditions when calculating the state of the chilling and forcing phases (Fu et al., 2012). The chilling phase starts at the onset of dormancy ($t_0$, fixed to 1 September), and is active on days with mean temperature between -5 and 10 °C. The rate of chilling can be expressed as follows:

$$R_C = \begin{cases} \dfrac{1}{1+e^{C_a(x-C_c)^2+C_b(x-C_c)}} & if -5 \leq x \leq 10 \\ 0 & if \ x > 10 \ or \ x < -5 \end{cases} \tag{1}$$

where $x$ is the daily mean air temperature, and $C_a$, $C_b$, and $C_c$ are chilling rate parameters.

The chilling phase ends with the onset of quiescence ($t_1$), which is when the state of chilling ($S_C$; Eq. 2) reaches the critical state of chilling (Eq. 3):

$$S_C = \sum_{t_0}^{t} R_C(x_t) \tag{2}$$

$$t = t_1 \ if \ S_C > C^* \tag{3}$$

The forcing phase begins at $t_1$, and it is active on days with mean temperature is above 0 °C.

$$R_F = \begin{cases} \dfrac{1}{1+e^{F_b(x-F_c)}} & if \ x > 0 \\ 0 & if \ x \leq 0 \end{cases} \tag{4}$$

where $F_b$ and $C_c$ are forcing rate parameters.





The forcing phase ends with the occurrence of SOS ($t_2$), which is when the state of forcing ($S_F$; Eq. 5) reaches the critical state of forcing (Eq. 6):

$$S_F = \sum_{t_1}^{t} R_F(x_t) \tag{5}$$

$$t = t_2 \quad if \quad S_F > F^* \tag{6}$$

The DM model (Delpierre et al., 2009) is based on the assumption that leaf senescence is driven both by the photoperiod and by low temperatures. This model represents the progress of senescence processes using a coloring state ($S_{sen}$) for each day ($d$), which is the accumulation of the daily rate of senescence ($R_{sen}$). Senescence starts on the date ($D_{start}$) when the photoperiod is shorter than a certain threshold ($P_{start}$), and the EOS is recognized as when $S_{sen}$ is larger than the threshold value $Y_{crit}$. The functions expressed in the process are as follows:

$$if \ P(d) < P_{start} \begin{cases} if \quad T(d) < T_b \quad R_{sen}(d) = [T_b - T(d)]^x \times f[P(d)]^y \\ if \quad T(d) \geq T_b \quad R_{sen}(d) = 0 \end{cases} \tag{7}$$

$$if \ \begin{cases} P(d) \geq P_{start} \quad S_{sen}(d) = 0 \\ P(d) < P_{start} \quad S_{sen}(d) = S_{sen}(d-1) + R_{sen}(d) \end{cases} \tag{8}$$

$$f[P(d)] = \frac{P(d)}{P_{strat}} \quad or \quad f[P(d)] = 1 - \frac{P(d)}{P_{strat}} \tag{9}$$

where $P(d)$ is the photoperiod on a given day $d$, $T(d)$ is the daily mean temperature (℃), and $Tb$ is the critical temperature below which senescence processes are effective (℃).

We calibrated and validated the two phenology models using all pixel-year SOS and EOS data extracted from the GIMMS3g NDVI dataset during 1982–2018. Five phenological extraction methods (i.e., the HANTS-Maximum, Spline-Midpoint, Gaussian-Midpoint, Timesat-SG, and Polyfit-Maximum methods) were applied to obtain the mean phenological index value, reducing the uncertainty caused by the use of only a single method (Fu et al., 2021). To match the spatial resolution of gridded meteorological data, the SOS and EOS data were resampled to 0.1° spatial resolution, and outliers were removed using the interquartile range method (Cui et al., 2017). We optimized parameters of the UniChill model (i.e., $C_a$, $C_b$, $C_c$, $F_b$, $C_c$, $C^*$, and $F^*$) and the DM model (i.e., $P_{start}$, $T_b$, $x$, $y$, and $Y_{crit}$) for each valid pixel in upper reaches of the Jinsha River watershed, minimizing root mean square error (RMSE) through particle swarm optimization. Phenology data from odd-numbered years were utilized for parameter optimization, and even-numbered years were used for validation.

The warming-up, calibration, and validation periods for the original and modified SWAT-Carbon models were set at 2005–2006, 2007–2011, and 2012–2014, respectively. The calibration of modified SWAT-Carbon model involved the following



two steps, whereas that of original SWAT-Carbon model only employed the second step. 1) We optimized the 8 parameters (Table S1) that control the shape, magnitude, and temporal dynamics of the LAI for forest and grassland through particle swarm optimization, to align the simulated LAI curves with the observed dynamic LAI. 10 particles with different initial velocities were generated through the Latin hypercube sampling method. The Nash–Sutcliffe efficiency (NSE) was used as the objective function to continuously update the parameters. Optimal parameter sets were determined when the objective function remained unchanged after 10 iterations. 2) We adjusted those parameters that control the hydrological processes using SWAT-CUP 2019 (https://swat.tamu.edu/software/swat-cup/) (Abbaspour, 2008). The Sequential Uncertainty Fitting (Sufi-2) algorithm in SWAT-CUP was used for sensitivity analysis, uncertainty analysis, and parameter calibration (Arnold et al., 2012). Overall, 18 parameters (Table S2) were used for model calibration. These parameters were derived by combining the 14 most sensitive parameters selected from the pool of 22 parameters for each original and modified model.

**2.4 Model Evaluation**

The widely used coefficient of determination ($R^2$), NSE, and percent bias (PBIAS) were selected as metrics to evaluate the performance of the model in simulating the LAI and runoff in our study area (Moriasi et al., 2007):

$$R^2 = \frac{\left(\sum_{i=1}^{n}\left(O_i - \overline{O}\right)\left(P_i - \overline{P}\right)\right)^2}{\sum_{i=1}^{n}\left(O_i - \overline{O}\right)^2 \sum_{i=1}^{n}\left(P_i - \overline{P}\right)^2} \tag{10}$$

$$NSE = 1 - \frac{\sum_{i=1}^{n}\left(O_i - P_i\right)^2}{\sum_{i=1}^{n}\left(O_i - \overline{O}\right)^2} \tag{11}$$

$$PBIAS = \frac{\overline{P} - \overline{O}}{\overline{O}} \times 100\% \tag{12}$$

where $n$ is the number of observations; $O_i$ and $P_i$ are the observed and simulated values, respectively, at time i; and $\overline{O}$ and $\overline{P}$ represent the average of the observed and simulated values, respectively.

**3 Results**

**3.1 Spatiotemporal Variations in Climate and Vegetation Phenology**

There is notable spatiotemporal heterogeneity in the multiyear mean annual temperature and accumulated precipitation across the upper reaches of the Jinsha River watershed (Figure 2a and b). The mean annual temperature exhibits a gradual increase from the upstream to the downstream areas. The annual accumulated precipitation is higher in the downstream





region compared to the upstream. In addition, the mean annual temperature and accumulated precipitation showed clear temporal trends during 1982-2018. The mean annual temperature significantly increased by 0.06 °C per year on average. The annual accumulated precipitation across the entire watershed increased by 2.64 mm per year, however, downstream regions exhibited a declining trend (Fig.2b). Over the past four decades, the upper reaches of the Jinsha River watershed have showed a "warming–wetting" trend in the upstream area and a "warming–drying" trend in some downstream areas.

During 1982–2018, the multiyear mean SOS and EOS in the upper reaches of the Jinsha River watershed were 164 ± 12.16 and 298 ± 10.56, (Day of year, DoY), respectively (Figure 2c and 2d). The mean SOS initiated in early May upstream and early July downstream. The mean EOS occurred in early October upstream, one month earlier than downstream. The SOS advanced in 60.5% of the study area over the past four decades, primarily in the middle and upper reaches of the watershed (-0.05 ± 0.13 days per year), while being delayed by 0.09 ± 0.30 days per year in the downstream area. During the study period, the EOS was delayed by 0.08 ± 0.27 days per year in the downstream area but advanced by -0.03 ± 0.14 days per year in the upstream area. This spatiotemporal heterogeneity of vegetation phenology further highlights the importance of incorporating vegetation dynamics into hydrological models.

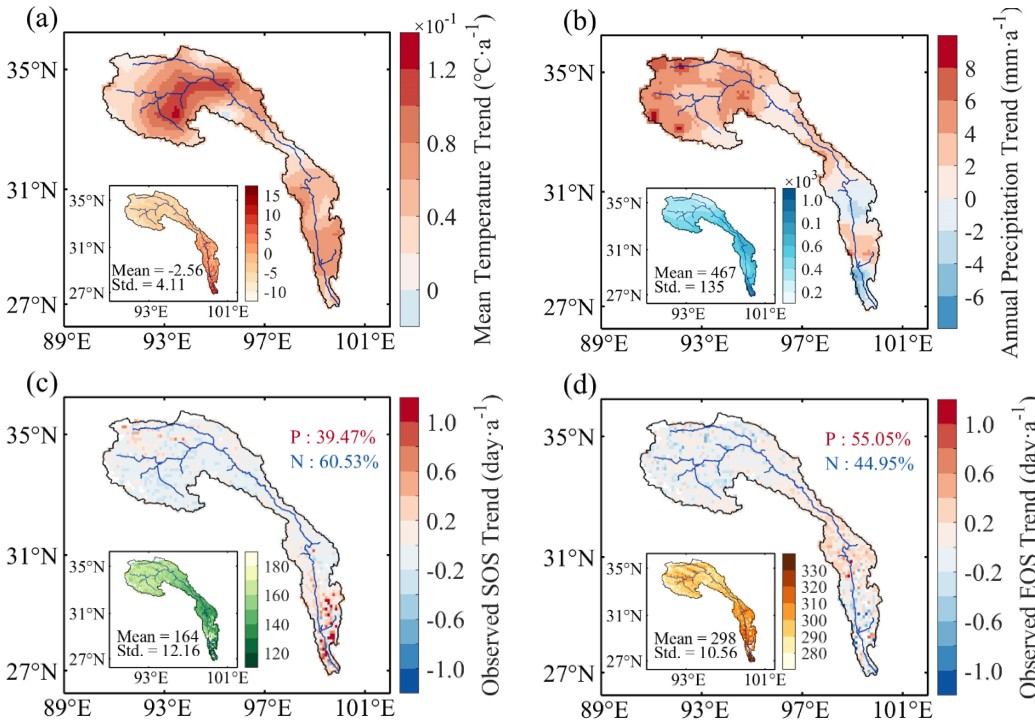

**Figure 2: Spatial and temporal variations of climatic variables and vegetation phenology during 1982–2018.** Temporal variations of mean annual temperature (a), accumulated annual precipitation (b), start-of-season (SOS, c) and end-of-season (EOS, d). The inner plot of each subfigure depicts the spatial pattern of the multi-year means of each variable.





## 3.2 Performance of Phenology Module and Modified SWAT-Carbon Model

The mean RMSE for the spring and autumn phenology models is 9.89 and 7.51 days, with calibration and validation RMSEs
of 8.71 and 10.82 days for SOS, and 7.03 and 7.77 days for EOS, respectively (Fig.3a and b). Within the study area, 67.1%
and 85.8% of pixels exhibited an RMSE of less than 10 days for SOS and EOS, respectively. The correlation coefficients
between observed and simulated SOS and EOS are 0.54 and 0.63, respectively.

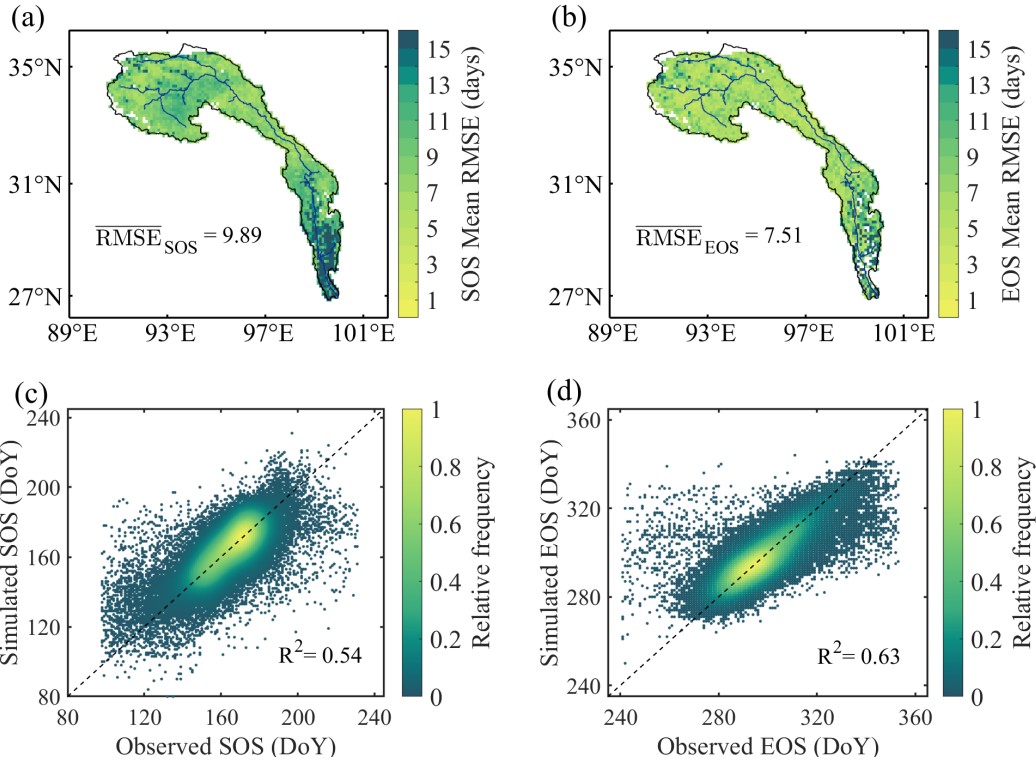

**Figure 3: Performance of phenology module.** Spatial patterns of the RMSE for start-of-season (SOS, a) and end-of-season (EOS, b).
Scatter plots depicting the relationship between simulated and observed start-of-season (SOS, c) and end-of-season (EOS, d). The colored
points on the scatter plots represent kernel density, with lighter colors indicating a higher density distribution. The dashed line represents
the 1:1 ratio line.

In the modified SWAT-Carbon model, the simulated LAI before the start of the growing season was changed to a non-zero
value for both the forest and grassland, and the calibrated and validated LAI growth curves largely improved compared to
the original model (Figure 4a and 4b, Table 2). Specifically, the average $R^2$ and NSE values improved from 0.31 and -1.42 to
0.79 and 0.78 for forest, and from 0.13 and -1.84 to 0.88 and 0.87 for grassland on average, respectively. Additionally, the
absolute PBIAS decreased by 41.0% for forest and 97.0% for grassland.

Incorporating phenology module into the SWAT-Carbon model enhanced the performance of runoff simulation. The
modified SWAT-Carbon model produced runoff simulations with a 'very good' level of performance ($0.80 < R^2 \leq 1$ and $0.75$





< NSE ≤ 1, Figure 4c and Table 2). Furthermore, the performance of runoff simulation in various months exhibited improvement, particularly during vegetation greening period (June), senescence period (October), and the non-growing season (Table S3). Specifically, the $R^2$ and NSE values for runoff increased by 0.13 and 0.39 in June, respectively, by 0.30 and 0.46 in October. The NSE value for runoff from November of the previous year to May increased by 0.35.

**Table 2: Evaluation indices of the original and modified SWAT-Carbon model in simulating LAI and runoff.**

| Variable | Evaluation indices | Calibration (2007–2011) | | Validation (2012–2014) | |
|---|---|---|---|---|---|
| | | Original | Modified | Original | Modified |
| LAI (forest) | $R^2$ | 0.28 | 0.77 | 0.33 | 0.80 |
| | NSE | -1.47 | 0.76 | -1.36 | 0.79 |
| | PBIAS (%) | 42.77 | -1.50 | 41.75 | 1.12 |
| LAI (grassland) | $R^2$ | 0.11 | 0.87 | 0.15 | 0.89 |
| | NSE | -1.87 | 0.86 | -1.80 | 0.88 |
| | PBIAS (%) | 97.47 | 0.65 | 97.15 | -0.01 |
| Runoff | $R^2$ | 0.93 | 0.95 | 0.89 | 0.92 |
| | NSE | 0.92 | 0.95 | 0.86 | 0.92 |
| | PBIAS (%) | 4.80 | 1.75 | 5.04 | 1.22 |

$R^2$, coefficient of determination; NSE, Nash–Sutcliffe efficiency; PBIAS, percent bias.

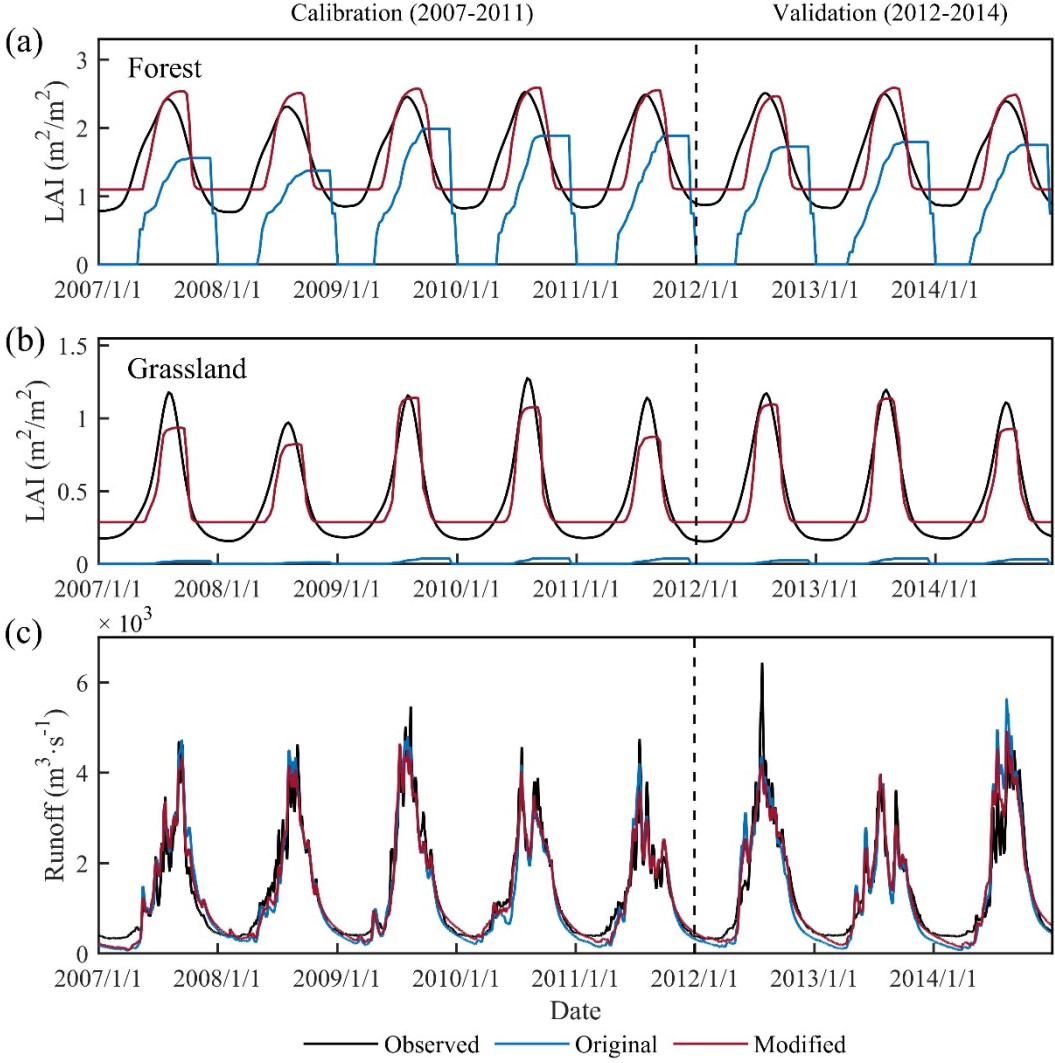

**Figure 4: Temporal variability of the LAI and runoff during the calibration (2007–2011) and validation (2012–2014) periods.** 8-d LAI time series observed by satellite and simulated by the original and modified SWAT-Carbon models for Forest (a) and grassland (b). Observed and simulated daily runoff by the original and modified SWAT-Carbon models (c).

### 235 3.3 Future Shifts in Vegetation Phenology and Runoff

The Unichill and DM phenology models were used to forecast future shifts in SOS and EOS during 2030–2100 under different emission scenarios. The SOS in the upper reaches of the Jinsha River watershed is projected to advance under all scenarios (Figure 5a), with the most significant advancement observed under the high-emission scenario (0.49 days per year, $p < 0.001$). In addition, a delaying trend in EOS was observed under all scenarios (Figure 5b), with the largest delay 240 occurring under the high-emission scenario (0.18 days per year, $p < 0.001$).

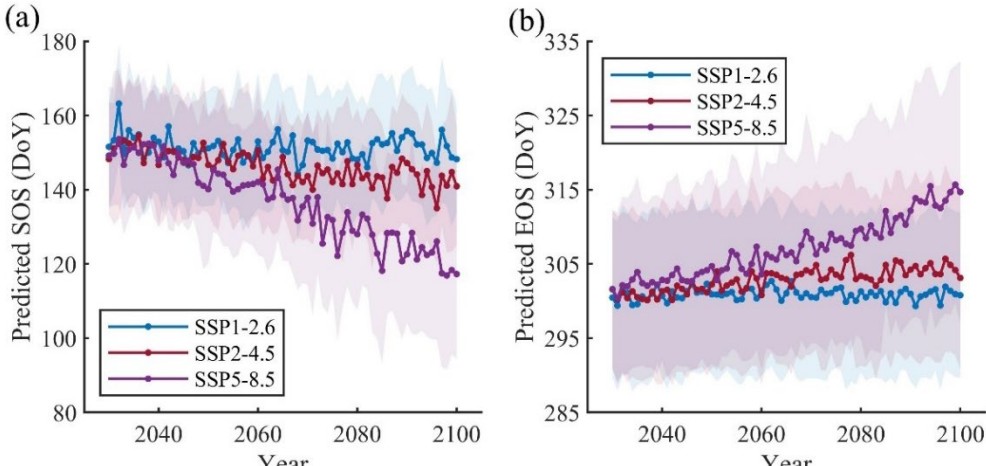

**Figure 5: Projection of future spring and autumn phenology during 2030–2100.** Future shifts in start-of season (SOS, a) and end-of-season (EOS, b) under three Shared Socioeconomic Pathway (SSP) scenarios based on the Coupled Model Intercomparison Project Phase 6 (CMIP6) multi-model ensemble. SSP1-2.6, SSP2-4.5 and SSP5-5.8 refer to low, moderate and high emissions, respectively. Colored
lines and shading represent the mean and one standard deviation, across the four CMIP6 models under each emission scenario.

The runoff under SSP5-5.8 exhibited greater fluctuation with continuous increase from 2023 to 2100. Specifically, runoff is projected to increase from approximately 1,437 m³ s⁻¹ in 2030 to 3,638 m³ s⁻¹ in 2100. However, the runoff under SSP1-2.6 and SSP2-4.5 is projected to initially increase and then stabilize after 2060 (Figure 6). The average runoff under SSP5-8.5 (2,459 m³ s⁻¹) is greater than that under SSP1-2.6 (2,180 m³ s⁻¹) and SSP2-4.5 (2,120 m³ s⁻¹). Results from the original
SWAT-Carbon model underestimated the future increases in runoff compared to those produced by the modified SWAT-Carbon model (Figure S1).

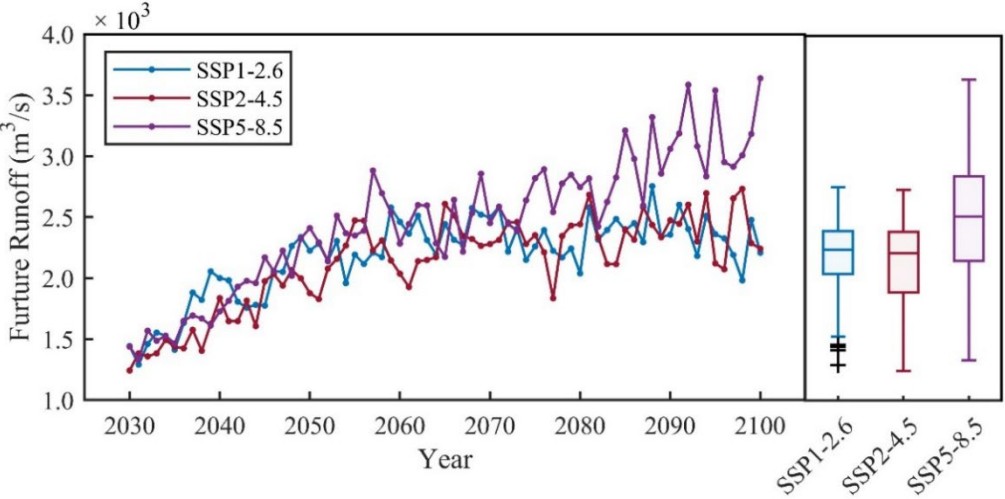

**Figure 6: Projection of future runoff during 2030–2100 using the modified SWAT-Carbon model.** The right subplot represents the annual mean runoff. SSP1-2.6, SSP2-4.5 and SSP5-5.8 refer to low emission, moderate and high emissions, respectively, based on the
Coupled Model Intercomparison Project Phase 6 (CMIP6) multi-model ensemble.





## 4 Discussion

### 4.1 Improvement of Runoff Simulation by Hydrological Model

Vegetation phenology has strong controls on seasonal and annual hydrological processes (Kim et al., 2018; Yang et al., 2023). During the start and end of the growing season, rapid changes in vegetation physiological properties such as stomatal

conductance and LAI influence the timing and amount of water resource allocation (Hwang et al., 2023; Zhang et al., 2021). Various environmental factors (e.g., temperature, precipitation, and radiation) and vegetation types determine the start and the end of the growing season (Wu et al., 2021), and process-based phenology models that incorporate different climatic factors have been well developed (Fu et al., 2020). However, the vegetation phenology in most hydrological models, such as the SWAT-Carbon model, is based on simple thermal and daylength thresholds (Arnold et al., 1998). This approach results

in larger uncertainty in vegetation dynamics and thereby hydrological processes predictions, especially under climate change (Zhang et al., 2009; Wang et al., 2022; Ma et al., 2019). In this study, we integrated process-based spring and autumn phenology models into the SWAT-Carbon model, which substantially improved the simulation of vegetation dynamics and ecohydrological processes, especially during the months of spring and autumn vegetation growth. Therefore, we highlight the importance of integrating a phenology module into hydrological models.

In addition to vegetation phenology, ecosystem structure, such as species composition is also affected by climate change (Chuine, 2010; Huang et al., 2019), which would substantially impact interactions between vegetation phenology and hydrology processes. Hence, more advanced land surface dynamic vegetation models, such as the LPJ-GUESS (Sitch et al., 2003), could be coupled with hydrological models to further improve our understanding of future vegetation dynamics and its effect on carbon and water cycles.

### 4.2 Interaction between Vegetation Phenology and Hydrological Processes

Vegetation phenology and hydrological processes are closely intertwined through biotic and abiotic pathways (Buermann et al., 2018; Lian et al., 2020). For example, the physiological activity of vegetation and its impact on the underlying surface properties of a watershed affect land surface evapotranspiration (Chen et al., 2022b). In line with previous studies in the Northern Hemisphere, our study revealed a positive correlation between ET and growing season length, potentially attributed

to the prolonged period of water movement from the soil to the atmosphere (Geng et al., 2020; Yang et al., 2023). This was further verified by the observation that earlier leaf unfolding and delayed leaf senescence increased spring and autumn ET (Figure S2). Furthermore, vegetation phenology influences the development and decay of the canopy, altering the water demands of vegetation, and contributing to the augmentation of the water cycle (Hwang et al., 2018; Lian et al., 2020). However, our study found that both spring and autumn ET increased by a similar magnitude, with values of 1.58 and 1.62

mm per day, respectively. This finding is inconsistent with the asymmetrical increase reported in previous research (Kim et





al., 2018), which could be attributable to the abundant soil moisture during summer in our study area. This moisture abundance does not impose constraints on vegetation growth and transpiration.

The unrealistic representation of vegetation phenological processes in current hydrological models often requires remediation through additional ecohydrological processes (Luan et al., 2022), thereby compromising the fidelity mechanics.
For instance, calibration algorithms often lead to inaccurate representation of ecological processes despite producing accurate runoff simulations. Therefore, improving the simulation accuracy of vegetation dynamics could enhance the capability of hydrological models to accurately depict ecohydrological processes.

### 4.3 Hydrological Response to Future Climate Change and Vegetation Dynamics

We predicted future runoff in the upper reaches of the Jinsha River watershed under and found that the runoff would largely
increase under future emission scenario. Under the SSP5-8.5 scenario, runoff exhibited the most pronounced upward trend, primarily attributed to increased precipitation largely surpassing that of the SSP1-2.6 and SSP2-4.5 scenarios. In addition, despite a substantial increase in precipitation under SSP2-4.5 compared to SSP1-2.6, the projected runoff under SSP2-4.5 is marginally smaller than that under SSP1-2.6. This phenomenon can be attributed to the extension of the growing season under global warming, which would significantly increase evapotranspiration under the moderate emission pathway
compared to the low emission pathway (Lu et al., 2021; Yang et al., 2023). The increased precipitation under SSP2-4.5 is insufficient to fully counteract the increase in evapotranspiration caused by the prolonged growing season (Figure S3).

In summary, our study integrated a process-based vegetation phenology module into the SWAT-Carbon model, substantially improving the simulation of vegetation dynamics and hydrological processes in the upper reaches of the Jinsha River watershed. The runoff is predicted to increase over 2030-2100 under different future emission scenarios, primarily due to
greater precipitation outweighing the increase in evapotranspiration induced by the prolonged growing season. Our study highlights the importance of integrating phenological shifts into hydrological models. In the future, it will be crucial to consider and couple plant functional types into hydrological models to enhance the performance of ecohydrological process simulations. Furthermore, the phenology module should be incorporated into other hydrological models and applied in various watersheds to enhance our understanding of ecohydrological processes and to support future water resources
management under climate change conditions.

### Data availability

Datasets for driving and evaluating the phenology models and the SWAT-Carbon model are available from the sources listed in Table 1.



**Author contributions**

Conceptualization: YF; Data curation: NW, JZ; Formal Analysis: ML, SC; Funding acquisition: YF, FH; Methodology: ML, SC, YX, ZW; Software: ML, SC; Supervision: YF; Validation: SC; Visualization: ML, JZ; Writing – original draft: ML; Writing – review & editing: SC, FH, NW, ZW, YX, JZ, YZ, YF.

**Competing interests**

The authors declare that they have no known competing financial interests or personal relationships that could have appeared 320 to influence the work reported in this paper.

**Acknowledgments**

This study was supported by the Joint Fund for Regional Innovation and Development of the National Natural Science Foundation of China (Grant No. U21A2039), the National Science Fund for Distinguished Young Scholars (42025101), and the National Natural Science Foundation of China (No.42330515).

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
