# Peer review of "Integration of the Vegetation Phenology Module Improves Ecohydrological Simulation by the SWAT-Carbon Model"

_Hydrology and Earth System Sciences, 2024_

## Author Comment (AC1)

**Response to Referee #1**

**[Comment 1]** The paper "Integration of the Vegetation Phenology Module Improves Ecohydrological Simulation by the SWAT-Carbon Model" modifies the SWAT-Carbon model to include a process-based method for estimating the start and end of the growing season based on parameterizations of environmental conditions. The paper is logically structured and provides an interesting and relevant analysis. The introduction provides a nice overview of previous research and sets up the study very nicely. The data and methods section describes the details of the experiment very well and is easy to understand. The results present an interesting analysis that compares the two model runs.

**[Response 1]** We thank the referee for the supportive comments. Please see below our responses to each comment.

**[Comment 2]** However, there are two main shortcomings of the paper. First, the paper would benefit from detailed revisions to address several grammatically incorrect and awkward sentences throughout the paper. This would greatly improve the readability and overall quality of the paper.

**[Response 2]** Following the referee's comment, we corrected the grammatical errors and improved the sentences by a professional expert. We are convinced that the readability of our manuscript has been greatly improved, please see the revised manuscript.

**[Comment 3]** Second, the discussion section is inherently weak and needs extensive revisions. There is little to no discission about the uncertainty of the underlying data sets and models and contextualizing this uncertainty within the analysis. There also seems to be little connection between the discussion and the results presented within the body of the paper. This makes it unclear as to what are the main results from the work. Overall, the paper is interesting and well suited for publication in HESS after some revisions to address these shortcomings. Some specific examples of ways to improve the manuscript are given below.

**[Response 3]** We thank for referee for these thoughtful comments and suggestions. In the revised manuscript, following the referee's suggestions, we substantially improved the discussion sections: (1) we revised the paragraph in section 4.1 in which we added the discussion of the potential uncertainty of datasets and models, please see the details in our response to similar comment#7; (2) we updated the discussion to strengthen the connection between the discussion and the results as the referee pointed out, please see the details in our responses to comments#7, 9 and 10

**Specific comments**
**[Comment 4]** Revisions: Line 123: This section needs a few sentences describing how "the current SWAT-Carbon model preforms poorly in estimating vegetation dynamics". This will then help justify why the authors decide to add the UniChill model and the DM model.

**[Response 4]** We thank the referee for this helpful comment. Following the referee's suggestion, we addressed the limitations of the original SWAT-Carbon model and explained our choice of the UniChill and DM models in the revised manuscript:

"*The current SWAT-Carbon model uses daylength thresholds, that are determined only by latitude, to simulate the onset and the end of vegetation dormancy. This approach fails to accurately capture vegetation dynamics as it largely ignores the effects of other important environmental variations (i.e., temperature) (Chen et al., 2023). Incorporating accurate phenology information could enhance the simulation of LAI, thereby improving the accuracy of runoff simulated by the hydrological model. The UniChill model and the DM model, which account for the response of phenology to various environmental variations, have been widely used to simulate spring and autumn phenology (Roberts et al., 2015; Yang et al., 2012). Therefore, we modified the SWAT-Carbon model by integrating the process-based vegetation phenology module.*"

**[Comment 5]** Line 218 – Changing the none growing season LAI to a non-zero value greatly improved the statistics, but this should have been done for the original model too to make it a fair comparison. As is, the statistical improvement between the original and the modified is mostly due to this arbitrary choice of changing the none growing season LAI to a non-zero value and does not capture the improvement in the model's ability to estimate SOS and EOS. The improvement from the two different changes in the modified model should be carefully analyzed and discussed.

**[Response 5]** We thank the referee for this thoughtful comment. Following the referee's suggestion, we revised the non-growing season LAI in the original SWAT-Carbon model by adjusting the default management operation schedule and values of minimum LAI parameters (ALAI_MIN). We found that compared to the default settings, there are only slight improvements in the LAI growth curves of the original model (Figure S1). Specifically, the average NSE value improved from -1.42 to -0.29 for forests and from -1.84 to -0.70 for grasslands, which is much smaller than NSE for LAI in the modified model (0.78 for forest and 0.87 for grassland). Furthermore, we even found that the average R² decreased by 0.17 for forests and by 0.01 for grasslands, respectively. These results demonstrated the importance of incorporating the phenological module into SWAT-Carbon model.

In the revised manuscript, we included these findings in the supplementary figure S1, along with additional descriptions of results and corresponding discussions:

"*To further exclude the impact of non-growing season LAI, we also changed the non-growing season LAI to non-zero in the original SWAT-Carbon model. The results indicated that, compared to the default settings, there are only slight improvements in the LAI growth curves in the original model (Figure S1). Specifically, the average NSE value improved from -1.42 to -0.29 for forests and from -1.84 to -0.70 for grasslands, which is much smaller than NSE for LAI in the modified model. Furthermore, the average R² even decreased by 0.17 for forests and by 0.01 for grasslands, respectively.*"

[Figure]

**Figure S1: Temporal variability of the LAI during the calibration (2007–2011) and validation (2012–2014) periods.** 8-d LAI time series observed by satellite and simulated by the original SWAT-Carbon model with adjusting non-growing season LAI for Forest (a) and grassland (b).

**[Comment 6]** Line 250: Be careful with the wording of "underestimated the future increases in runoff". You cannot "underestimate" something that is not known. It would be better to say "the original SWAT-Carbon model shows a smaller increase in future runoff compared to the modified SWAT-Carbon". Then this can be followed up with a discussion about why the original SWAT carbon underestimated historical data and what that may mean about the future.

**[Response 6]** Following the referee's suggestion, we revised relevant expression in the revised manuscript as: "*Results from the original SWAT-Carbon model showed a smaller increase in future runoff compared to those estimated by the modified SWAT-Carbon model*".

In addition, we analyzed the water components in both the original and modified SWAT-Carbon models to investigate the underlying reasons for the historical runoff underestimation by the original model. The potential reason is that the low baseflow in the original model, attributed to unrealistic LAI simulations and unreasonable parameters, led to the underestimation of runoff simulation, particularly during the non-rainy season. Following the referee's suggestion, we added the discussion about the potential implications of this underestimation for future projections:

"*In this study, future runoff simulation of original SWAT-Carbon model is smaller than that of modified model, especially under the high emission scenario, which is also observed in the historical runoff (Figure 4c). The underestimation of historical runoff simulation is attributed to lower baseflow compared to observed data, particularly during the non-rainy season. This discrepancy is potentially influenced by underestimated LAI simulations and unreasonable parameters in original SWAT-Carbon model (Tang et al., 2022). Vegetation degradation is reported to lead to decreased flow during the dry season (Paiva et al., 2023). In addition, although*

*calibration algorithms enhance the original model's ability to produce accurate runoff simulations, the unrealistic representation of vegetation phenological processes necessitates remediation through additional ecohydrological processes (Luan et al., 2022; Haas et al., 2022), thereby compromising the model's fidelity. Therefore, improving the simulation accuracy of vegetation dynamics could enhance the capability of hydrological models to accurately depict ecohydrological processes, especially in the dry season, thereby facilitating more effective water resource conservation strategies under future global warming."*

**[Comment 7** Line 258: The discussion in section 4.1 is inherently weak. Discussion should focus on contextualizing the results from the study with broader results in the research. Yet, there is little discussion about the results from this paper (lines 266-268) and it mostly reads like an introduction section that provides broad background. The paper would be improved by starting this section about what improvements this work demonstrated and then discussing what they mean in a broader context as well as potential sources of uncertainty.

**[Response 7]** Following referee's suggestion, we largely expanded the discussion about improvements of the modified SWAT-Carbon model as:

*"In this study, we integrated process-based spring and autumn phenology models into the SWAT-Carbon model, which substantially improved the simulation of vegetation dynamics and ecohydrological processes. The simulation of the LAI curve, particularly in terms of the seasonality and magnitude, exhibited significant improvement in modified SWAT-Carbon model (Figure 4a and 4b, Table 2). The inaccuracy of LAI simulation in the original model is attributed to the initiation and end of vegetation dormancy being determined only by a latitude-based daylength threshold. This approach leads to mismatches in the timing of phenological stages in the LAI curve compared to observed data (Figure 4a, 4b and Figure S1). Based on the widely used processed phenological model (Fu et al., 2020), we incorporated the phenology module into the SWAT-Carbon model, which provides a method for accurately simulating the timing of vegetation growth stages and the runoff simulation. In addition, the largest and lowest simulated LAI values in original model showed significant underestimation. This is attributed to the inability to meet the default constant value of heat accumulation requirements, owing to low daily temperatures in the upper reach of Jinsha River watershed. In the modified model, we therefore replaced the constant value with a dynamic heat accumulation requirement, and optimized vegetation growth parameters, which finally enhanced the simulation accuracy of LAI values and thereby improved runoff simulation. Therefore, we highlight the importance of integrating a phenology module into hydrological models."*

In addition, we added the discussion about potential sources of uncertainty:

*"Our study indicated that the modified SWAT-Carbon model improved the simulation of vegetation and runoff. However, some uncertainties from dataset and model persist in the modified model should be addressed. The meteorological input data used in this*

*study were obtained from a gridded source, which may differ from actual conditions. In addition, climate-sensitive ecosystem structures, such as species composition, introduces uncertainties in assessing interactions between vegetation phenology and hydrological processes in the modified SWAT-Carbon model (Chuine, 2010; Huang et al., 2019). Therefore, coupling advanced land surface dynamic vegetation models, such as LPJ-GUESS (Sitch et al., 2003), with hydrological models could further improve our understanding of future vegetation dynamics and their effects on the carbon and water cycles.*"

**[Comment 8]** Line 275: This discussion section also needs revisions. When does the analysis show "a positive correlation between ET and growing season length"? If the figures in supplementary material are so important that they are worth discussing in the discussion section, then they should probably be included in the paper

**[Response 8]** We thank the referee for this comment. Following referee's suggestion, we incorporated the figure S2 into the section 4.2 of the paper and updated the corresponding text in the revised manuscript:

"*In addition, the vegetation dynamics and its impact on the underlying land surface properties affect watershed evapotranspiration (Chen et al., 2022b). To explore the response of ET to vegetation phenology in the upper reach of Jinsha River watershed, we analyzed the relationship between the phenological variations and sub-basin scales ET using modified SWAT-Carbon model. Consistent with previous studies in the Northern Hemisphere, our study revealed a positive correlation between ET and growing season length (Figure 7), potentially attributed to the prolonged period of water movement from the soil to the atmosphere (Geng et al., 2020; Yang et al., 2023).*"

[Figure]

**Figure 7: Relationship between the phenological variations and sub-basin scales evapotranspiration (ET) using modified SWAT-Carbon model.** GSL, growing season length; SOS, start-of-season; EOS, end-of-season; ET, evapotranspiration.

**[Comment 9]** Again, like the last section, it is difficult to parse out what the authors are discussing as a main finding in the work as most of the discussion section is broad and is only loosely connected with the results.

**[Response 9]** Following the referee's suggestion, in the revised manuscript we updated this section as:

"*Our results indicated significant improvement in the performance of runoff simulation, particularly during the vegetation greening period (June) and the senescence period (October) (Table 2 and Table S3), which is attributed to the accurate phenology prediction. Vegetation phenology and hydrological processes are closely intertwined through biotic and abiotic pathways (Buermann et al., 2018; Lian et al., 2020). In the upper reach of Jinsha River watershed, the multiyear mean SOS and EOS occurred in June and October (Figure 2), respectively. During the start and end of the growing season, rapid changes in vegetation physiological properties such as stomatal conductance and LAI influence the timing and amount of water resource allocation*

*(Hwang et al., 2023; Zhang et al., 2021).”*

**[Comment 10]** Line 293 – Again, in this section the authors are primarily focusing on results from the supplementary material (Figure S3) and do not reference much of the results in the paper. This should be better aligned with the main message of the paper and connect the discussion to the main findings.

**[Response 10]** Following referee's suggestion, we expanded the discussion aligned with the main findings:

*"We predicted future runoff in the upper reaches of the Jinsha River watershed and found that the runoff would largely increase under future climate change conditions. Specifically, under the SSP5-8.5 scenario, runoff exhibited the most pronounced upward trend, primarily attributed to increased precipitation largely surpassing that of the SSP1-2.6 and SSP2-4.5 scenarios (Figure 6 and Figure S3). In addition to the precipitation, the early SOS and delayed EOS under global warming (Figure 5), which were predicted in Jinsha River watershed, also play an important role in altering the water cycle. Despite a substantial increase in precipitation under SSP2-4.5 compared to SSP1-2.6, the projected runoff under SSP2-4.5 is marginally smaller than that under SSP1-2.6. This phenomenon is likely attributed to the extension of the growing season under global warming, which would significantly increase evapotranspiration under the moderate emission pathway compared to the low emission pathway (Lu et al., 2021; Yang et al., 2023)."*

**[Comment 11]** Line 302 – This paragraph while under the heading of section 4.3, is more general than just section 4.3 so should maybe be part of a new section 5 conclusions where the broader results of the work are summarized. This paragraph is also redundant with other parts of the discussion sections.

**[Response 11]** We thank the referee for this helpful comment and agree that the paragraph would be more appropriately positioned in a new Section 5. Following the referee's suggestion, we relocated this paragraph to Section 5 in the revised manuscript.

**Additional references cited in our response to Referee #1 as:**

Haas, H., Kalin, L., and Srivastava, P.: Improved forest dynamics leads to better hydrological predictions in watershed modeling, Sci. Total Environ., 821, 153180, https://doi.org/10.1016/j.scitotenv.2022.153180, 2022.

Paiva K., Rau P., Montesinos C., Lavado-Casimiro W., Bourrel L., and Frappart F.: Hydrological Response Assessment of Land Cover Change in a Peruvian Amazonian Basin Impacted by Deforestation Using the SWAT Model, Remote Sens., 15, 5774, https://doi.org/10.3390/rs15245774, 2023.

Roberts, A. M. I., Tansey, C., Smithers, R. J., and Phillimore, A. B.: Predicting a change in the order of spring phenology in temperate forests, Glob. Change Biol., 21, 2603–2611, https://doi.org/10.1111/gcb.12896, 2015.

Tang Z., Zhou Z., Wang D., Luo F., Bai J., and Fu Y.: Impact of vegetation restoration on ecosystem services in the Loess plateau, a case study in the Jinghe Watershed, China, Ecol. Indic., 142, 109183, https://doi.org/10.1016/j.ecolind.2022.109183, 2022.

Yang X., Mustard J. F., Tang J., and Xu H.: Regional-scale phenology modeling based on meteorological records and remote sensing observations, J. Geophys. Res. Biogeo., 117, https://doi.org/10.1029/2012JG001977, 2012.

---

## Author Comment (AC2)

**Response to Referee #2**

**[Comment 1]** This study developed a process-based vegetation phenology module and coupled it with the SWAT-Carbon model. This modified model demonstrates improved performance in simulating both vegetation dynamics and runoff in the upper reaches of the Jinsha River watershed, and it is applied to investigate the vegetation effects on runoff. This study shows the strong influence of vegetation phenology on hydrological processes and highlights the importance of integrating a phenology module into hydrological models. The manuscript is well-written and improves the SWAT-Carbon model for historical and future ecohydrological simulation under climate change, though some details need to be further explained and modified. Some detailed suggestions and comments are listed below:

**[Response 1]** We thank the referee for the supportive and constructive comments. Please find below our point-by-point response to each comment raised.

**[Comment 2]** Introduction: The manuscript does not emphasize the significance of choosing the Jinsha River watershed as the study area for applying the modified model. It would be better to clarify the importance of this study area in light of its ecohydrological characteristics.

**[Response 2]** Following the referee's comment, we emphasized the significance of choosing the upper reaches of the Jinsha River watershed as our study area in the revised manuscript as: "*The upper reaches of the Jinsha River watershed, originating from the Tibetan Plateau and forming the upper reach of Yangtze River, is recognized as one of the ecologically fragile regions in China. In recent years, the upper reaches of the Jinsha River watershed experienced seriously climate change, greatly influencing vegetation dynamics and regional water cycles (Wu et al., 2020; Jiang et al., 2022; Li et al., 2022). Enhancing our understanding and simulation of ecohydrological processes in the upper reaches of the Jinsha River watershed is critical for securing water resources and maintaining ecological stability.*"

**[Comment 3]** Line 100: Why are these four CMIP6 models chosen for prediction? Some models, such as CanESMs, have coarse spatial resolution especially when applied at the watershed scale. The authors should clarify the considerations for their selection of these climate models.

**[Response 3]** We thank the referee for this suggestion. We selected the four CMIP6 models based on two main criteria. The first is that these models provide daily time outputs of mean temperature, maximum temperature, minimum temperature, and precipitation, which were used to drive phenology models and SWAT-Carbon model. The second is that these models were commonly used in studies of Yangtze River (Zhang et al., 2023; He et al., 2024). To reduce the effect of resolution differences, we downscaled the spatial resolution of climate data from the four CMIP6 models to match the observation data, and further corrected systematic biases using the empirical quantile mapping technique.

Following the referee's comment, we clarified this in the method section of the manuscript as: "*CMIP6 outputs were used to predict vegetation phenology and future runoff. We selected four CMIP6 models, i.e., CanESM5, FGOALS-g3, MPI-ESM1-2-HR, and MRI-ESM2-0, which were commonly used in studies of Yangtze River (Zhang et al., 2023; He et al., 2024). The daily time series of mean temperature, maximum temperature, minimum temperature, and precipitation under three emission scenarios (SSP1-2.6, SSP2-4.5, and SSP5-8.5) were acquired from the CMIP6 website (https://esgf-node.llnl.gov/search/cmip6/).*"

**[Comment 4]** Line 250: Future runoff changes simulated by the modified SWAT-Carbon model could be compared not only with those simulated by the original SWAT-Carbon model but also with runoff directly provided by CMIP6.

**[Response 4]** Following the referee's comment, we added a comparison of future runoff simulated by the modified SWAT-Carbon model with those directly provided by CMIP6. We found that the average future runoff obtained from CMIP6 models were larger than those simulated by the modified SWAT-Carbon model (please see the Fig. S3 below and the revised manuscript). In the revised manuscript, we incorporated these findings in the results as:

"*Monthly total runoff outputs were obtained from CMIP6 models and aggregated to annual runoff from 2030 to 2100. We calculated the watershed-scale future runoff from CMIP6 using the area-weighted mean method and found that the future runoff from CMIP6 models were larger than that simulated by the modified SWAT-Carbon model (Figure S3).*"

[Figure]

**Figure S3: Projection of average future runoff from 2030 to 2100 under each emission scenario simulated by the modified SWAT-Carbon model and CMIP6.**

**[Comment 5]** Figure 2: The information (e.g., abbreviations) in the figures should be clearly interpreted in the figure caption.

**[Response 5]** Following the referee's comment, we improved the caption of Figure 2.

[Figure]

**Figure 2: Spatial and temporal variations of climatic variables and vegetation phenology during 1982–2018.** Temporal variations of mean annual temperature (a), accumulated annual precipitation (b), start-of-season (SOS, c) and end-of-season (EOS, d). The inner plot of each subfigure depicts the spatial pattern of the multi-year means of each variable. P, percentage of pixels showing a positive trend, indicates increased temperature and precipitation, or delayed SOS and EOS; N, percentage of pixels showing a negative trend, indicates decreased temperature and precipitation, or advanced SOS and EOS.

**[Comment 6]** Figure 6: These times series should include shading to represent the uncertainty, as shown in Figure 5?

**[Response 6]** Following the referee's comment, we incorporated shading into Figure 6 to represent the uncertainty across the four CMIP6 models under each emission scenario.

[Figure]

**Figure 6: Projection of future runoff during 2030–2100 using the modified SWAT-Carbon model.** Colored lines and shading in the right subplot represent the mean and one standard deviation across the four Coupled Model Intercomparison Project Phase 6 (CMIP6) models. The scenarios SSP1-2.6, SSP2-4.5 and SSP5-5.8 refer to low emission, moderate and high emissions, respectively.

**[Comment 7]** Abbreviations in the text need to be consistent to enhance readability.

**[Response 7]** Following the referee's comment, we reviewed and corrected abbreviations throughout the manuscript to ensure consistency.

**Additional references cited in our response to Referee #2 as:**

He, K., Chen, X., Zhou, J., Zhao, D., and Yu, X.: Compound successive dry-hot and wet extremes in China with global warming and urbanization, J. Hydrol., 636, 131332, https://doi.org/10.1016/j.jhydrol.2024.131332, 2024.

Jiang, Q., Yuan, Z., Yin, J., Yao, M., Qin, T., Lü, X., and Wu, G.: Response of vegetation phenology to climate factors in the source region of the Yangtze and Yellow Rivers, J. Plant Ecol., 17, rtae046, https://doi.org/10.1093/jpe/rtae046, 2024.

Wu Y., Fang H., Huang L., and Ouyang W.: Changing runoff due to temperature and precipitation variations in the dammed Jinsha River, J. Hydrol., 582, 124500, https://doi.org/10.1016/j.jhydrol.2019.124500, 2020.

Zhang, C., Sun, F., Sharma, S., Zeng, P., Mejia, A., Lyu, Y., Gao, J., Zhou, R., and Che, Y.: Projecting multi-attribute flood regime changes for the Yangtze River basin, J. Hydrol., 617, 128846, https://doi.org/10.1016/j.jhydrol.2022.128846, 2023.